# The Impact of the COVID-19 Pandemic on Mental Health and Substance Use among People with and without HIV

**DOI:** 10.3390/pathogens12030461

**Published:** 2023-03-15

**Authors:** Morgan Zabel, Tony W. Wilson, Harlan Sayles, Pamela E. May, Renae Furl, Sara H. Bares

**Affiliations:** 1College of Medicine, University of Nebraska Medical Center, Omaha, NE 68198, USA; 2Institute for Human Neuroscience, Boys Town National Research Hospital, Omaha, NE 68010, USA; 3College of Public Health, University of Nebraska Medical Center, Omaha, NE 68198, USA

**Keywords:** COVID-19, mental health, substance use, HIV

## Abstract

People with HIV (PWH) may be particularly vulnerable to the psychological impacts of COVID-19. To assess this, participants were recruited from two established cohorts of PWH and HIV− adults with the available pre-pandemic baseline data and completed the Beck Depression Inventory-II (BDI-II), Beck Anxiety Inventory (BAI), Alcohol Use Identification Test (AUDIT), National Institute on Drug Abuse Quick Screen (NIDA-QS), and Pittsburgh Sleep Quality Index (PSQI) at two distinct intra-pandemic time periods. All outcomes were evaluated using generalized linear mixed models. In total, 87 participants completed all the questionnaires; 45 were PWH and 42 were HIV−. The pre-pandemic mean BDI-II, BAI, AUDIT and PSQI scores were higher in the PWH cohort. After the onset of the pandemic, the mean BDI-II, AUDIT and PSQI scores increased within the sample as a whole (*p* < 0.001, *p* = 0.029 and *p* = 0.046, respectively). The intra-pandemic mean BDI-II scores fell slightly for both groups and the AUDIT scores increased slightly for the PWH group and fell slightly for the HIV− group, but not significantly. The intra-pandemic PSQI scores increased sharply for both groups. The percentage of PWH and HIV− participants who moved into a more severe category of depression was identical (18%), but more PWH met the criteria for clinical evaluation. The BAI and NIDA-QS scores did not increase significantly. In conclusion, the measures of mental health symptoms and alcohol use increased in both groups after the onset of the pandemic. Although there were no significant differences in the changes between the groups, the PWH had higher baseline scores and the changes in this group had more clinical impacts.

## 1. Introduction

The COVID-19 pandemic and the associated public health measures introduced to mitigate the spread of the virus have had significant impacts on mental health and substance use. One survey of adults living in the United States (US) found a 3-fold increase in the prevalence of anxiety symptoms, a 4-fold increase in the prevalence of depressive symptoms and a 2-fold increase in suicidal ideation after the onset of the pandemic [1]. In the same survey, 13% of adults reported starting or increasing substance use to cope with stress or emotions related to COVID-19.

People with HIV (PWH) may be particularly vulnerable to the psychological impacts of COVID-19 as they were disproportionately impacted by stigma, discrimination, and isolation before the onset of the pandemic and inevitably experienced further social isolation as a result of the measures put in place to minimize the spread of the virus [2]. In addition, PWH experience higher rates of cardiovascular disease, diabetes, and chronic kidney disease, which may place them at higher risk for severe COVID-19 and, thus, more liable to experience fear and anxiety at the outset of the pandemic [3,4]. Further, PWH have higher baseline rates of mental health, alcohol and substance use disorders than people without HIV and these could be further exacerbated by the pandemic [5,6].

At the same time, PWH can be a resilient population [7] and resilience has been noted to be a protective factor against psychological distress in adverse situations [8]. For example, a large survey study of people with and without HIV found that PWH had lower odds of anxiety and higher odds of resilience than people without HIV [9]. Further, anxiety was associated with a higher risk of substance use and resilience was associated with a lower risk of anxiety and substance use.

However, several recent cross-sectional studies have found PWH to have an increase in mental health symptoms during the COVID-19 pandemic [10], including depressive symptoms [11,12], anxiety [12,13,14], and insomnia [12]. While studies have evaluated differences in mental health and substance use disorder in patients with and without HIV, no studies to date have done so with both pre- and intra-pandemic data and over two distinct intra-pandemic time periods.

Having a better understanding of how the pandemic has affected PWH over these time periods is critical to inform our screening policies for mental health symptoms and substance use and to identify the priority areas for scarce mental health resources during a national shortage of mental health providers.

To evaluate both the short- and long-term impact of the COVID-19 pandemic on mental health and patterns of substance use among people with and without HIV, we conducted a longitudinal cohort survey study and compared the results collected at two distinct time periods during the pandemic to pre-pandemic data.

## 2. Materials and Method

### 2.1. Study Design and Participants

Participants were recruited from two established cohorts of PWH and HIV− adults in Omaha, Nebraska for whom baseline data, including mental health and substance use assessments, had been collected prior to the pandemic (Multimodal Imaging of NeuroHIV Dynamics (R01-MH118013) and Signatures of Cannabis Abuse in NeuroHIV (R01-DA047828). Participants in the parent studies were people with and without HIV who were over the age of 19 years. PWH were required to be taking combination antiretroviral therapy (ART) and have an HIV−1 RNA of less than 20 copies/mL at the time of enrollment. Participants were excluded from the parent studies if they had any known psychiatric or neurological diagnoses (with the exception of HIV-associated cognitive impairment) or were taking antipsychotics, anticonvulsants, or other major psychotropic medications.

A hybrid of telephone verbal and online remote informed consent was obtained from all participants prior to enrollment. All questionnaires were completed electronically via the Collaborative Informatics and Neuroimaging Suite (COINS) software [15,16] during two different periods spaced approximately 6 months apart (visit 1 was between 15 February and 26 April 2021 and visit 2 was between 15 August and 29 November 2021). To avoid duplicate surveys, all participants were assigned a unique identification number and an individualized link to the study questionnaires. Upon completion of the questionnaires, participants with severe mental health symptoms (BDI-II score of 21 or greater or item 9 greater than 0, or BAI score of 22 or greater) were provided with immediate mental health resources and referrals.

Demographics were collected at baseline as were medical histories and medication lists and the latter were updated at visit 1 and visit 2. Questionnaires completed pre-pandemic (as part of the parent studies) and then again at visit 1 and visit 2 included: Beck Depression Inventory-II (BDI-II) [17], Beck Anxiety Inventory (BAI) [18], Alcohol Use Identification Test (AUDIT) [19], National Institute on Drug Abuse Quick Screen (NIDA-QS) [20], and Pittsburgh Sleep Quality Index (PSQI) [21]. A questionnaire regarding COVID-19-related stressors and secondary adversities was collected at visit 1 and visit 2. Of note, the BAI was scored on 17 items rather than 21 as the pre-pandemic surveys were collected on paper and not all participants completed the last four items.

### 2.2. Ethics Approval

The study protocol was approved by the University of Nebraska Medical Center’s Institutional Review Board and participation in the study was voluntary.

### 2.3. Statistical Analysis

Participant characteristics were summarized by HIV status. Differences in subject characteristics and COVID-19 survey responses between groups were evaluated using t-tests (with a Satterthwaite approximation for unequal variances as needed) for continuous measures and exact Pearson chi-square tests for categorical measures. Simple associations between outcome measures at each visit were assessed using Pearson correlation coefficients. All outcomes were evaluated using generalized linear mixed models and all outcomes were evaluated overall and by HIV status. For overall models, the models contained fixed effects for visit (baseline or visit 2 vs. visit 1), subject age, and subject sex and a random effect for subject. Visit 1 was selected as the reference category as most comparisons of interest involved comparing the early pandemic against either the time before the pandemic or the late pandemic. For models by HIV status, fixed effects for group and a group-by-visit interaction were also included. In all models AUDIT, BAI, BDI-II, and PSQI counts were modeled using Poisson distributions while positive screenings for NIDA-QS measures were modeled using binomial distributions. For the NIDA-QS measures regarding frequency of use, anything other than “Never” was counted as a positive screening. All analyses were conducted using SAS version 9.4.

## 3. Results

### 3.1. Cohort Characteristics

A total of 87 participants completed all questionnaires at baseline, visit 1 and visit 2; 45 were PWH and 42 were HIV negative controls (HIV−). Their characteristics by group are presented in Table 1. The participants with HIV tended to have lower levels of education and the PWH had significantly lower incomes than the HIV− participants as a percentage of the federal poverty level in 2021 but not 2019.

### 3.2. Depressive and Anxiety Symptoms

Pre-pandemic mean BDI-II and BAI scores were somewhat, though not significantly, higher in the PWH versus the HIV− participants. After the onset of the pandemic, the mean BDI-II scores increased significantly in both groups at visit 1 and remained elevated at visit 2 (Figure 1, *p* < 0.001). The percentage of the PWH and HIV− participants who moved into a more severe category of depression as measured by the BDI-II (e.g., from minimal to moderate) after the onset of the pandemic was identical (18%). The mean BAI scores increased numerically at the start of the pandemic and then decreased to below baseline levels intra-pandemic, but not significantly (Figure 2). When compared to the HIV− cohort, at the start of the pandemic, the mean BDI-II scores increased slightly less and the BAI scores increased slightly more among the PWH, but not significantly. Of note, eleven participants (eight PWH and three HIV−) met the criteria for clinical intervention based on high BDI-II and BAI scores (BDI-II ≥ 22, BDI item 9 > 0, and/or BAI ≥ 22) and were immediately contacted for further evaluation.

### 3.3. Alcohol Use

Pre-pandemic mean AUDIT scores were higher in the PWH group versus the HIV− group, though not significantly. After the onset of the pandemic, the mean AUDIT scores increased significantly among both groups (Figure 3; *p* = 0.029) and remained elevated at visit 2. The AUDIT and NIDA-QS alcohol screening scores were significantly higher in males than females (*p* = 0.001 and *p* = 0.31, respectively). Changes in AUDIT scores were higher, but not significantly so, among the cohort of PWH.

### 3.4. Other Substance Use

The use of illegal drugs as measured by the NIDA-QS was trending lower at visit 1 relative to baseline and had decreased significantly relative to baseline by visit 2 (coefficient −1.79; *p* = 0.026); there was no difference between the PWH and HIV− cohorts.

### 3.5. Sleep

Overall, the mean PSQI scores were initially higher, indicating worse sleep, in the PWH cohort versus the HIV− cohort (*p* = 0.045). The mean PSQI scores increased significantly in both groups at visit 1 (*p* = 0.046) and were even more elevated by visit 2 (*p* < 0.001). There was no significant difference in the rate of increase between groups. High PSQI scores significantly correlated with both high levels of anxiety (BAI) and depression (BDI-II) scores among the sample as a whole and at all visits (*p* = 0.001 or smaller for all).

### 3.6. Indirect Impacts of COVID-19 Questionnaire

Both groups reported increased financial hardship due to COVID-19 at visit 1, however the PWH reported significantly more financial hardships versus the HIV− group, 38% and 12%, respectively (*p* = 0.007). By the second visit, only 29% and 12% of the PWH and HIV− groups, respectively, were reporting a financial hardship and the difference was no longer significant, although it was close (*p* = 0.066). At visit 1, most participants (78% of the PWH cohort and 88% of the HIV− cohort) reported they stopped seeing their friends and loved ones in person due to COVID-19, but those numbers were somewhat reduced (53% of the PWH cohort and 74% of the HIV− cohort) by visit 2. Just over half of the participants in each cohort (52% of the PWH cohort and 56% of the HIV− cohort) reported experiencing feelings of loneliness over the prior 6 months at visit 1 and these numbers remained steady at visit 2 (49% of the PWH cohort and 57% HIV− cohort).

## 4. Discussion

We conducted a longitudinal cohort survey study to assess the symptoms of mental health and patterns of substance use among people with and without HIV before and during the COVID-19 pandemic. In this analysis, we found that both the PWH and HIV− groups experienced increased depressive symptoms one year into the pandemic, compared to baseline. Although PWH had higher baseline BDI-II scores, the percentage of the PWH and HIV− participants who moved into a more severe category of depression (e.g., from minimal to moderate) after the onset of the pandemic was identical (18%). That said, there were more PWH who met our pre-defined criteria for further mental health evaluation and support so the increases in depressive symptoms among PWH necessitated more clinical interventions and mental health referrals. It is also worth emphasizing the BDI-II trends over the course of the study. The distribution of the intra-pandemic BDI-II scores in the HIV− group resembled the pre-pandemic distribution in the PWH group. This subtle, yet important finding yields several points of discussion. Baseline depression scores in the PWH cohort were similar to the intra-pandemic scores in the HIV− group and this resemblance may be attributable to the mental health impacts of living with a chronic disease. The stressors of facing a stigmatized health condition and the resulting psychosocial impacts are not unfamiliar to PWH. In other words, PWH may have been experiencing the challenges of an epidemic long before COVID-19 happened. In light of these results, PWH may be considered a priority group during the mental health professional shortage.

Although anxiety as measured by the BAI did not increase significantly in either group, the PWH experienced more sleep difficulties at baseline, and this correlated with anxiety and depressive symptoms. Anxiety symptoms may not have increased in our study due to the BAI only including 17 out of the 21 items, leading to falsely reassuring results. Of note, the last few items in the BAI are more somatic in nature, such as experiencing indigestion/abdominal discomfort, fainting, face flushing, and sweating not due to heat and could be related to other underlying medical conditions and not solely anxiety. Prior studies have shown that PWH experienced less symptoms of anxiety than HIV− groups during the pandemic and resiliency has been noted to be a protective factor [22].

Previous pre-pandemic studies have demonstrated that PWH, in particular males, have high rates of alcohol use disorder (AUD) [6]. Our results are consistent with these prior findings and are notable given the increase in alcohol use among both cohorts after the onset of the pandemic and the persistent nature of the change, with a sustained increase at visit 2 which occurred over 1 year into the pandemic. These results highlight the need for more frequent and consistent screening for alcohol use, especially for males, given the enduring nature of the increase in alcohol use. These results are contrasted with a previously mentioned study of people with and without HIV in Miami, Florida, where they did not observe significant changes in alcohol consumption but found alcohol use was more prevalent among the HIV− participants [22]. Of note, PWH in this study reported less symptoms of anxiety and higher levels of resilience than the HIV− cohort. Additionally, this study only looked at intra-pandemic data and did not compare alcohol use patterns to pre-pandemic data.

In contrast to other studies, we found that substance use decreased after the onset of the pandemic. While this data is reassuring, prior studies have demonstrated increases in substance use during the pandemic ranging from 3.6% to 17.5%, with mental health factors being the most common correlates or triggers [23]. While unmeasured resilience may have played a role in the decrease in reported substance use among the cohort of PWH, it is also possible that our cohort is not representative of the general population, given participants were excluded from the parent studies if they used substances other than marijuana.

It is also worth noting that our study may have selected out a highly resilient group of PWH given that the parent studies required participants to have an undetectable HIV RNA at entry. Similarly, just by participating in the study, the PWH group demonstrated a level of intrinsic motivation, likely translating to a certain degree of resiliency which would be a protective factor for mental health symptoms and substance use in the pandemic.

HIV is a disease that is embedded in structural inequities as it affects those of lower socioeconomic status at a disproportionately high rate [24], and the same has been noted for the COVID-19 pandemic [25]. The results in our study continue to support this, as the PWH reported significantly more financial hardships after the onset of the pandemic than the HIV− group. Although these groups did not show significant differences in income pre-COVID (measured as a percentage of the federal poverty limit), incomes were inversely impacted by 2021 with the PWH having less income, and the HIV− participants having more income. This widening difference between the groups is notable given lower education levels among the PWH group and demonstrates increased vulnerability to the economic impacts of the COVID-19 pandemic.

There are several strengths and limitations to our findings. As mentioned above, it is important to note that the participants in the PWH cohort were required to have undetectable viral loads upon entry to the parent studies and this may have resulted in a selection bias of a group of patients that is highly resourceful and resilient, which are protective factors against substance use and mental health symptoms. Further, this study excluded participants with pre-pandemic regular substance use, other than marijuana, as it was an exclusion criterion in the parent studies. That we found significant changes in mental health symptoms despite these entry criteria, however, is remarkable. Additionally, as the study is based on self-reported online questionnaires, it is possible mental health symptoms and substances were underreported. Additionally, anxiety symptoms may have been underreported as the last few questions of the BAI were not collected at baseline and therefore not evaluable for comparison. Lastly, although not different between the two cohorts, most of the participants were White males living in Nebraska and therefore, these results may not be generalizable to the entire U.S. or global population. The strengths of our study include the detailed collection of a breadth of data for two well-established cohorts of PWH and HIV− groups at three distinct time points, including baseline pre-pandemic as well as two distinct timepoints during the pandemic.

## 5. Conclusions

Measures of depression and alcohol use increased significantly and remained elevated over 18 months after the onset of the COVID-19 pandemic in participants with and without HIV. Although there were no significant differences in the changes between the groups, PWH had higher baseline scores and the changes in this group prompted more clinical interventions. Further, PWH were more likely to report experiencing increased financial hardships and sleep difficulties.

Screening for symptoms of mental health and substance use is critical during a pandemic, especially in PWH, and, together with increased availability of mental health services, would form part of a robust pandemic preparedness package. Future studies will investigate the correlation between pre- and intra-pandemic mental health symptoms and substance use and their clinical impacts in PWH, such as virologic suppression, medication adherence, and retention in care.

## Figures and Tables

**Figure 1 pathogens-12-00461-f001:**
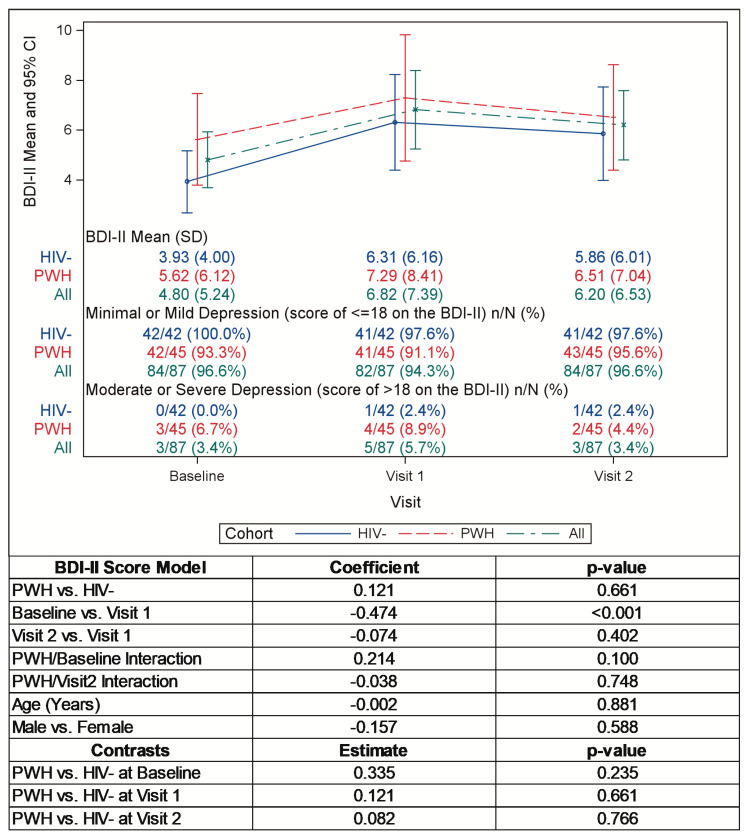
Mean BDI-II scores pre- and intra-pandemic.

**Figure 2 pathogens-12-00461-f002:**
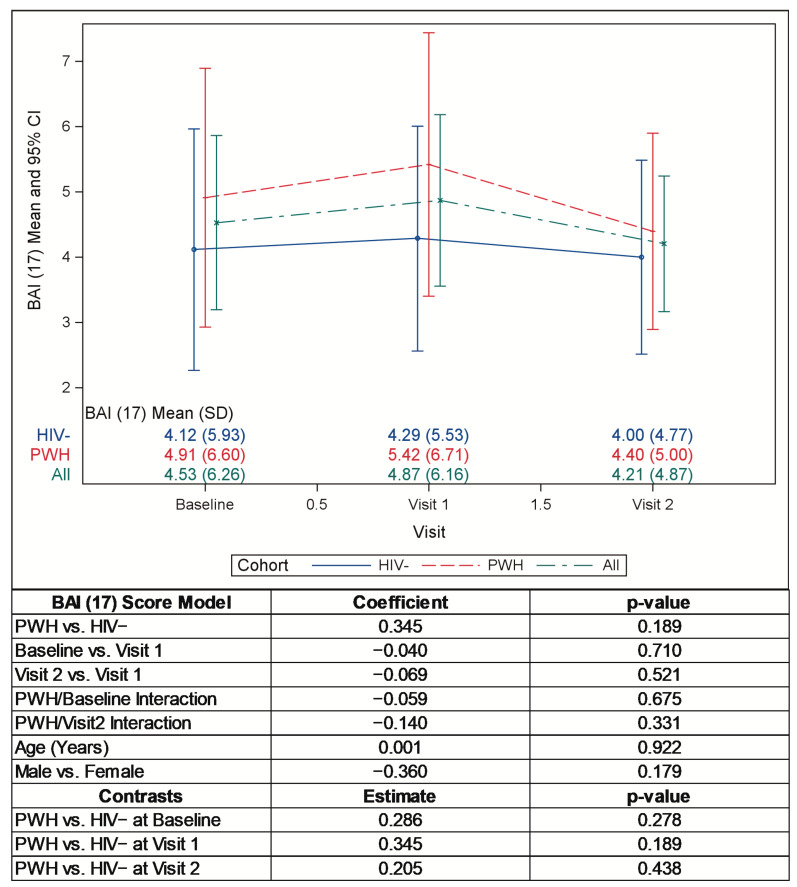
Mean BAI (17) scores pre- and intra-pandemic.

**Figure 3 pathogens-12-00461-f003:**
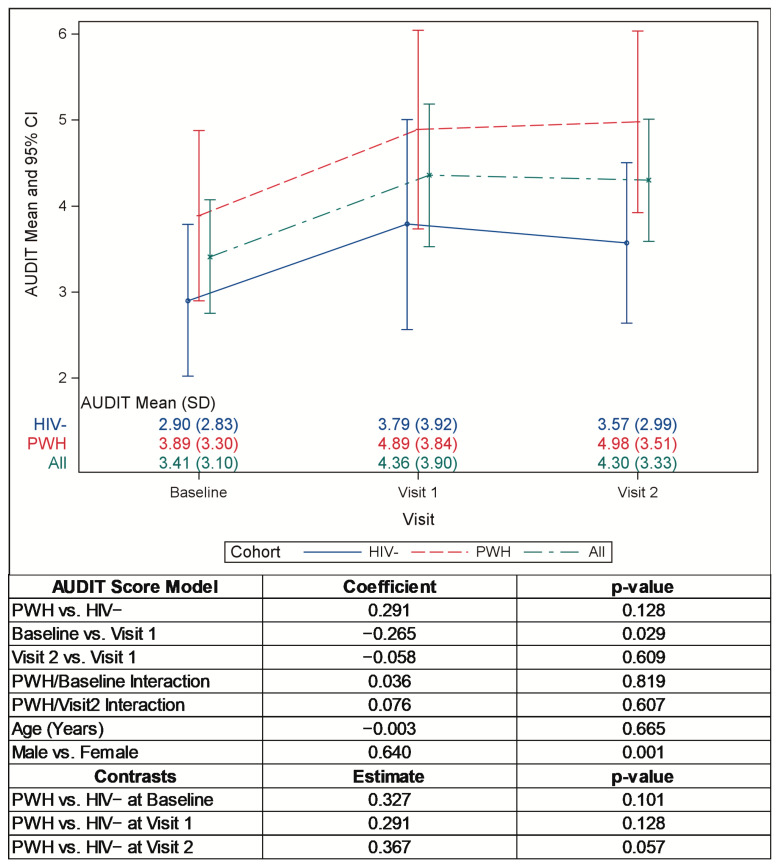
Mean AUDIT scores pre- and intra-pandemic.

**Table 1 pathogens-12-00461-t001:** Participant characteristics.

	HIV− (*N* = 42)	PWH (*N* = 45)	*p*-Value
Age at Baseline (years), mean (sd)	47.1 (13.4)	45.2 (13.0)	0.492
Age at Baseline, *n* (%)			0.584
21–30	6 (14)	10 (22)	
31–40	8 (19)	5 (11)	
41–50	8 (19)	11 (24)	
51–68	20 (48)	19 (42)	
Sex, *n* (%)			0.82
Male	27 (64)	31 (69)	
Female	15 (36)	14 (31)	
Race and Ethnicity, *n* (%)			0.689
Non-Hispanic White	33 (79)	32 (71)	
Non-Hispanic Black	4 (10)	8 (18)	
Hispanic	1 (2)	2 (4)	
Other	4 (10)	3 (7)	
Highest Level of Education, *n* (%)			0.001
Some grade/high school	0 (0)	5 (11)	
High school graduate	0 (0)	7 (16)	
Some college	7 (17)	16 (36)	
Associate degree	5 (12)	4 (9)	
Bachelor’s degree or greater	30 (71)	13 (29)	
Income, *n* (%)			0.251
Less than $25,000	4 (10)	11 (25)	
$25,000–50,000	7 (17)	6 (14)	
Over $50,000, under $100,000	8 (19)	7 (16)	
$100,000 plus	14 (33)	8 (18)	
Prefer not to answer	9 (21)	12 (27)	
Federal Poverty Level 2019, mean (sd)	470 (314)	344 (266)	0.086
Federal Poverty Level 2021, mean (sd)	494 (334)	319 (226)	0.008
Essential Worker, *n* (%)			0.286
Yes	19 (45)	26 (58)	
No	23 (55)	19 (42)	
Number residing in house, *n* (%)			0.319
1	11 (26)	15 (33)	
2	15 (36)	21 (47)	
3	9 (21)	5 (11)	
≥4	7 (17)	4 (9)	

## Data Availability

The data presented in this study are available on request from the corresponding author. The data are not publicly available due to privacy concerns.

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
