# Peer review of "The Impact of the COVID-19 Pandemic on Mental Health and Substance Use among People with and without HIV"

_pathogens, 2023, doi:10.3390/pathogens12030461_

Round 1

Reviewer 1 Report

In the current study, the authors investigated the psychological impact of COVID-19 in individuals with and without HIV. The study is well-designed and the manuscript is well-written. However, I have some reservations about the manuscript in its present form, which are detailed below.

1. Introduction

The authors postulate that PHW are more vulnerable to the psychological effects of COVID-19. The underlying theoretical rationale for this is not made clear. Especially since they describe PHW being more resilient than people without HIW. Is their hypothesis based on a minority stress model?

2. Statistical analysis

To analyze the data the authors used general lineal models. Why did they run two separate models (overall model and a model by HIV status)? A general linear model with group as between-subject factor and time with three levels as within-subject factor could also be done to evaluate main and interaction effects. Alternatively, baseline values could also be entered as a covariate to control for baseline effects.

3. Results

The numbers in figure 1 (96.6% moderate to severe depression at baseline seems very high. Please check the scores in figure 1)

Reviewer 2 Report

This manuscript by Zabel M investigates to evaluate both the short- and long-term impact of the COVID-19 pandemic on mental health and patterns of substance use among people with and without HIV.

Key findings

a.     measures of mental health symptoms and alcohol use were increased in both groups (with and without HIV) after the onset of the pandemic.

b.     There is no significance different between HIV positive and negative. However, People living with HIV had higher baseline scores.

First of all, congratulations to all the authors for presenting such an important topic to the scientific community.

This manuscript is well written. However, there are a few minor concerns

1.     In the table 1. Under the race and ethnicity percentage (in HIV-), the value doesn’t add to 100. Please revisit.

2.     Again in table 1, Under the highest level of education (in HIV+), the value doesn’t add to 100. Please revisit.

Please revisit all the percentage calculations.

Thanks

Reviewer 3 Report

Thank you for this interesting paper providing some useful insights into the impact of the COVID-19 pandemic on people living with and without HIV.

The paper is generally well written and structured, and I have only minor comments for each section below. 

INTRODUCTION

This section provides a helpful background and justification for your study. This study is especially interesting because it compares two cohorts (HIV positive and negative).

One minor recommendation on line 42: I would suggest, ‘At the same time, PWH can be a resilient population...’ 

MATERIALS AND METHODS

This section provides a detailed overview of your methodology and how you gathered and analysed the data. I note that you had ethical approval for the study was granted, and you obtained consent from participants.

I also note that participants with severe symptoms were referred for additional support. 

RESULTS

These provide some interesting insights into the impact of the COVID-19 pandemic on the two cohorts. It’s useful to have pre-pandemic data, to compare with data gathered during visits 1 and 2 (both in 2021, and therefore over a year into the pandemic). What’s especially interesting are similarities in some of the measures between groups – suggesting that pandemic experience can be universal regardless of pre-existing variables (e.g., living with HIV or not). 

It's also interesting to see the changes between visit 1 and visit 2 – especially around seeing friends, and financial hardship. It would be useful to know (or speculate) why this may be the case, given that this is a time slot in the middle of the pandemic. Also, I would be interested to see more examples of gender differences, which are only noted about alcohol use. Can any other variables be disaggregated (though I appreciate the relatively low number of female participants)?

DISCUSSION

You draw on your findings and link this other studies. I appreciate that you highlight structural inequities with relation to both HIV and COVID-19, and the study does provide useful confirmation. 

CONCLUSION

This is useful and includes helpful recommendations for increased focus on screening for mental health and substance use during a pandemic. I’d recommend adding that this would form part of a robust pandemic preparedness package; one of the lessons learned from COVID-19. 

REVIEWER RECOMMENDATIONS 

1.     Minor word changes on line 42.

2.     Explore further gender disaggregation if the data allow.

3.     Consider adding reference to pandemic preparedness in the conclusion.

Round 2

Reviewer 1 Report

The authors have addressed all comments.